# An Ex Vivo Intervertebral Disc Slice Culture Model for Studying Disc Degeneration and Immune Cell Interactions

**DOI:** 10.3390/cells14161230

**Published:** 2025-08-08

**Authors:** Eunha G. Oh, Li Xiao, Zhiwen Xu, Yuan Xing, Yi Zhang, Parastoo Anbaei, Jialun A. Chi, Li Jin, Rebecca R. Pompano, Xudong Li

**Affiliations:** 1Department of Orthopaedic Surgery, University of Virginia, Charlottesville, VA 22908, USA; eo7gj@virginia.edu (E.G.O.); lixiao.uva@gmail.com (L.X.); zx8uvc@virginia.edu (Z.X.); yx6b@virginia.edu (Y.X.); 218202117@csu.edu.cn (Y.Z.); pkp4qq@virginia.edu (J.A.C.); lj7q@virginia.edu (L.J.); 2Department of Spine Surgery, Spinal Deformity Center, The Second Xiangya Hospital of Central South University, Changsha 410011, China; 3Department of Chemistry, University of Virginia, Charlottesville, VA 22908, USA; pa5ze@virginia.edu (P.A.); rrp2z@virginia.edu (R.R.P.); 4Department of Biomedical Engineering, University of Virginia, Charlottesville, VA 22908, USA

**Keywords:** intervertebral discs, disc degeneration, tissue slice, macrophages, back pain

## Abstract

Intervertebral disc degeneration is a leading cause of back and leg pain and a major contributor to disability worldwide. Despite its prevalence, treatments remain limited due to incomplete understanding of its pathology. In vivo models pose challenges for controlled conditions, while in vitro cell cultures lack key cell–cell and cell–matrix interactions. To address these limitations, we developed a novel tissue slice culture model of mouse discs, in which intact mouse discs were sliced down to 300 μm thickness with a vibratome and cultured ex vivo at various time points. The cell viability, matrix components, structure integrity, inflammatory responses, and macrophage interactions were evaluated with biochemistry, gene expression, histology, and 3D imaging analyses. Disc slices maintained structural integrity and cell viability, with preserved extracellular matrix in the annulus fibrosus (AF) and mild degeneration in nucleus pulposus (NP) by day 5. Interleukin-1 (IL-1) induced disc degeneration manifested by increased glycosaminoglycan release in media and reduced *aggrecan* and *collagen II* mRNA levels in disc cells. Cultured disc slices promoted macrophages towards pro-inflammatory phenotype with elevated mRNA levels of *il-1α*, *il-6*, and *inos*. Macrophage overlay and 3D imaging demonstrated macrophage infiltration into the NP and AF tissues up to ~100 µm in depth. The disc tissue slice model captures key features of intervertebral discs and can be used for investigating mechanisms of disc degeneration and therapeutic evaluation.

## 1. Introduction

Disc degeneration is a complex, multifactorial process involving biochemical, genetic, mechanical, and environmental factors [1]. Disc degeneration is a key contributor to back pain and disability worldwide, affecting millions of individuals and incurring approximately $40 billion annually in the United States [2]. Despite its high prevalence, the molecular mechanisms of disc degeneration are not fully understood, and effective treatments for the prevention or reversal of degeneration are lacking [3,4].

Inflammation is recognized as a key driver in the progression of disc degeneration and associated pain. During disc degeneration, structural disruption permits macrophage infiltration, worsening the microenvironment and accelerating the process. Degenerated disc cells release chemokines and cytokines that recruit macrophages [5]. Elevated macrophage infiltration has been linked to disease severity, and both human and animal studies have demonstrated heterogeneous macrophage phenotypes [6]. We have previously demonstrated a peak in macrophage infiltration at disc hernia sites on postoperative day (POD) 7 in a mouse model, with these macrophages likely derived from circulating monocytes [7,8,9]. Similarly, Nakawaki and Kawakubo et al. showed that macrophages infiltrate the intervertebral discs as early as POD 1 in mice, with a rapid surge of pro-inflammatory M1 macrophages in the earlier phase [6,10,11,12]. Both Ling and Nakazawa et al. reported a significant increase in M1 macrophages in damaged disc regions in the earlier phase of human degenerated disc tissues [13,14]. In contrast, Djuric et al. demonstrated that anti-inflammatory M2 markers (CD163 and CD209) surpassed the pro-inflammatory M1 marker CD192 in disc samples from patients with symptomatic lumbar and cervical disc herniation [15]. These results suggest a heterogeneous population of macrophages is involved in the pathogenesis of disc degeneration, although their roles remain elusive.

To elucidate the role of macrophages in disc degeneration, in vitro models have been developed by co-culturing macrophages and disc cells. For example, studies showed that conditioned media from M1-polarized macrophages inhibited nucleus pulposus (NP) cell proliferation and promoted their catabolic processes, whereas media from M2-polarized macrophages supported NP cell proliferation and promoted anabolic activities [10,16,17,18]. In another study, Lu et al. demonstrated that overexpressing High Mobility Group Box 1 (HMGB1) in macrophages accelerates disc degeneration via NF-κB activation and MMP3 upregulation in NP cells, and that treatment with glycyrrhizin, an HMGB1 inhibitor, mitigated these effects [19]. Similarly, the secreted factors from inflammatory discs induced macrophage migration and activation [16]. However, these in vitro 2D monolayer culture models do not maintain cell–matrix and matrix–matrix interactions, often suffering from phenotypic changes that may not reflect the complex in vivo conditions. In vivo models, while valuable for studying the complex interactions within the disc microenvironment, pose significant challenges for controlled conditions. The highly dynamic nature of in vivo systems makes it difficult to capture temporal and spatial changes during disease progression. Additionally, the intricate cell–cell, cell–matrix, and biochemical signals complicate the ability to isolate the contribution of single parameters. These limitations highlight the need for a model that bridges the gap between in vivo and in vitro approaches.

Ex vivo tissue slice culture addresses these challenges by preserving the native tissue architecture and cellular microenvironment, enabling controlled experiments on cellular behavior and extracellular matrix (ECM) remodeling in a physiologically relevant setting. Tissue slices retain key organ-specific features, including metabolic activity, tissue homeostasis, and aspects of immune function, making them a powerful tool for studying complex biological processes [20,21]. This approach has been widely utilized in areas like neuroscience [22], pancreas [23], liver [24], cancer [25], and lymph nodes [26], enabling detailed investigations of disease mechanisms and therapeutic responses within these organs and tissues.

In this study, we developed a novel disc tissue slice culture system to investigate the process of disc degeneration and the crosstalk between macrophages and degenerated discs. After optimizing the thickness of tissue slices, we assessed the cell viability and structure of disc slices over time. Next, we validated the disc cell metabolism in response to inflammatory stimuli. Finally, we utilized this platform to investigate the crosstalk between macrophages and disc tissue slices. Our findings indicate that this ex vivo disc tissue slice model maintains key features of disc structure and function, offering a valuable platform for studying the underlying mechanisms and evaluating potential therapeutic interventions for disc degeneration and associated back pain.

## 2. Materials and Methods

### 2.1. Mouse Lumbar Disc Isolation and Disc Slicing

All animal procedures were approved by the Institution Review Board following the animal use guidelines. Lumbar spines were dissected from both male and female C57BL/6J mice (10–16 weeks of age) following euthanasia in a CO_2_ chamber per our established protocols [27]. Discs were isolated from young, healthy mice to establish a baseline of structural and cellular integrity, allowing for the controlled induction and study of early degenerative events. The intact lumbar discs with cartilage endplates were carefully dissected from lumbar spines and subjected to sequential immersions in PBS followed by Dulbecco’s Modified Eagle Medium/Nutrient Mixture F-12 (DMEM/F12, Thermo Fisher Scientific, Waltham, MA, USA), both supplemented with 2% penicillin–streptomycin.

Under sterile conditions, low melting point agarose (6in PBS, pH 7.4) was poured into a 35 mm Petri dish. The discs were axially embedded into the bottom of the dish before agarose started to solidify. An 8 mm tissue punch (World Precision Instruments, Sarasota, FL, USA) was used to extract a circular agarose slab containing the embedded disc. The agarose slabs were glued upside down to the stage with Duro Super Glue (Cyanoacrylate) and then submerged in ice-cold 1× PBS. A vibratome (Leica VT 1000S, Bannockburn, IL, USA) at a speed setting of 70 (0.13 mm/s) and frequency of 3 (30 Hz) was used to section disc slices with a thickness of 300 μm. Each whole disc yielded two quality disc slices.

Due to the jelly-like nature of NP tissue, the disc slices were easily damaged during transfer and culture. To prevent NP loss, they were transferred to a 150 mm Petri dish and re-encapsulated with 3% agarose. A thin layer of agarose was pipetted onto one side of the disc slice to cover the exposed NP region. Once the agarose solidified, the slice was carefully flipped to apply agarose to the opposite side. Excess agarose surrounding the tissue was removed using a 3–4 mm smaller tissue punch. Subsequently, the slices were placed in a cell culture plate containing complete growth medium (DMEM-F12 with 10% Fetal Bovine Serum (FBS) and 2% Antimycotic-Antibiotic solution (anti-anti)). Disc slices were cultured in a standard cell culture incubator at 37 °C and 5% CO2, with 200 μL of media in a 48-well plate. The parameters were applied in most experiments. The structural integrity of the disc slices was assessed using bright-field microscopy with a Axiovert 40C microscope (Zeiss microscope, Oberkochen, Germany).

### 2.2. IL-1β Treatment of Disc Slices

Disc slices were cultured in a 24-well plate with complete growth media (400 µL per well) for 1 day and then treated with various doses of recombinant human IL-1β (Biolegend, San Diego, CA, USA), at 0, 0.01, 0.1, and 1 ng/mL for an additional 24 h. Disc slices and culture media were harvested for RNA isolation and biochemical assays, respectively.

### 2.3. Live Dead Assay to Assess Disc Cell Viability

Fresh disc slices were cultured in a 48-well plate in complete growth medium (200 μL per well) for various time points (day 0, 1, 3, and 5), and cell viability was evaluated with a Live/Dead Reduced Biohazard Viability/Cytotoxicity Kit (Thermo Fisher Scientific, Waltham, MA, USA) following the manufacturer’s instruction. In brief, disc slices were incubated with the dyes for 2 h at room temperature, imaged using a Zeiss AxioZoom.V16 microscope (Carl Zeiss Microscopy, White Plains, NY, USA). Imaging was performed at × 40 magnification to capture the entire axial disc structure, and NP and AF were captured at × 112 magnification for enlarged details. Live (green) and dead (red) cells in all regions were quantified using ImageJ (v1.53, NIH, Bethesda, MD, USA) to calculate the percentage of viable cells in the NP and AF regions at various time points.

### 2.4. Histological Analysis

To assess disc structure and extracellular matrix, disc slices were fixed with 4% Paraformaldehyde (PFA) for 24 h, embedded in O.C.T Compound (Thermo Fisher Scientific, Waltham, MA, USA), and cryosectioned at 10 μm thickness. The Safranin-O/Fast Green staining and Alcian Blue/Picrosirius Red (ABPR) staining were performed following our previously established protocol to assess the proteoglycan and collagen contents [28]. Disc histology was analyzed by two independent researchers based on the disc scoring system previously defined in the literature [7,8,27].

### 2.5. Polarized Light Microscopy (PLM) Imaging and Analysis

To evaluate collagen fibers (birefringent under polarized light) in disc slices over time, we performed PLM imaging on Alcian Blue Picrosirius Red (ABPR) stained sections on a Nikon ECLIPSE E600 microscope (Nikon Instruments Inc, Melville, NY, USA) with Nikon Elements software (v4.60). Disc tissues were sectioned at a thickness of 10 μm. Images were acquired at ×200 magnification at consistent exposure settings across all samples to ensure comparability. Regions of interest (ROIs) were defined as the central AF region. For quantitative analysis, the entire field of view was used as the ROI to ensure consistency across samples. Birefringent collagen fibers were captured in colors ranging from green (indicating loosely packed, less mature collagen) to red and orange hues (indicative of more mature, highly cross-linked collagen). For quantitative analysis, PLM images were processed using ImageJ. Each image was split into red and green channels, and the mean pixel intensity for each color was measured within defined regions of interest (ROIs). The green-to-red (G/R) ratio was calculated for each image to provide a relative index of collagen fiber change over time, with higher G/R values indicating a greater proportion of immature collagen fibers [29,30].

### 2.6. TUNEL Assay for Cell Apoptosis

To evaluate the presence of apoptotic cells in the cultured disc slices, terminal deoxynucleotidyl transferase-mediated d-UTP Nick End Labeling (TUNEL) assay was performed with an in situ Cell Death Detection Kit (Roche, Basel, Switzerland). The cryostat disc slice sections were equilibrated to room temperature, permeabilized with 0.1% Triton X-100 in PBS, and then incubated with TUNEL reaction mixture for 60 min. They were mounted with Prolong Gold anti-fade reagent with DAPI (Life Technologies, Carlsbad, CA, USA). Fluorescence signals were visualized using a Nikon ECLIPSE E600 fluorescent microscope with Nikon Elements software. The TUNEL index was determined as the percentage of TUNEL-positive cells relative to the total cell counts in each tissue section. For each time point (0, 1, 3, and 5 days), 4–6 sections were analyzed, with disc slices from at least four independent experiments.

### 2.7. Biochemical Assays

The released sulfated glycosaminoglycan (GAG) was measured to determine the proteoglycan content in culture media by a dimethyl methylene blue (DMMB) colorimetric assay using chondroitin sulfate as a reference [7,8,31]. The optical density (O.D.) was recorded using the Spectramax ABS plus plate reader (Molecular Devices, Sillicon Valley, CA, USA) at 560 nm.

The concentration of nitric oxide in the culture media was measured using a Griess assay kit (Promega, Madison, WI, USA) according to established protocols [32].

Lactate dehydrogenase (LDH) is an intracellular enzyme found in most cells and is commonly used as a biomarker of cell damage or cytotoxicity in cell culture. LDH assay was performed per manufacturer’s guideline (Sigma-Aldrich, Saint Louis, MO, USA).

### 2.8. Macrophages and Disc Slice Co-Culture

Raw 267.4 macrophages (ATCC, Manassas, VA, USA) were cultured in DMEM-high glucose supplemented with 10% FBS, 1% penicillin (100 μg/mL), and 1% streptomycin (100 μg/mL) (Thermo Fisher Scientific, Waltham, MA, USA). Macrophages were seeded in a 48-well plate at a density of 2 × 10^5^/mL (300 μL per well). After one day, macrophages were switched to serum-free medium and cultured for 1 day under three conditions: with four disc slices (equivalent to two native discs), without disc slices, or with LPS (10 ng/mL, positive control). To facilitate direct macrophage interaction, tissue slices used in the overlay experiments were not capped with agarose on the top surface. Cells were imaged with a Carl Ziess Microscope (Carl Zeiss Microscopy, White Plains, NY, USA) at a × 200 magnification. Cell morphology parameters such as cell area, perimeter, and aspect ratio were measured using ImageJ. Cells and media were harvested for RNA isolation and biochemical assay, respectively.

### 2.9. Macrophage Overlay Experiment

Disc slices were cultured in a 48-well plate in complete growth media (DMEM-F12, 10% FBS, 2% Anti-Anti) (200 µL per well) for 3 days. RAW 267.4 macrophages (10^6^/mL) were labeled with CellTracker™ CM-DiI Dye (Thermo Fisher Scientific, Waltham, MA, USA) per the manufacturer’s instructions. Log growth macrophages were trypsinized and spun down at 1500 rpm for 5 min and then resuspended in 20 μL of fresh medium. DiI-macrophages (~10 µL, approximately 100,000 cells) were then overlaid onto the top of the disc slices, and cultured for 2 h at 37 °C. Then, additional serum-free media (200 μL) were gently added to the surrounding tissue in each well. After 24 h, the disc slices were rinsed three times with a large volume of PBS to remove floating and loosely attached cells, fixed in 4% PFA for 1 h, and then imaged with a Nikon A1Rsi confocal upright microscope (Nikon Instruments Inc, Melville, NY, USA) at ×40 and ×600 magnifications. Confocal z-stack images of the AF and NP were captured at 10 μm intervals, followed by three-dimensional reconstruction using NIS-Elements software. Whole discs served as controls. ImageJ was used to quantify penetration depth and cell counts at each depth.

### 2.10. RNA Isolation and RT-PCR

Disc slices were homogenized in a bead tube with 500 μL Trizol using the Fisherbrand^TM^ Bead Mill 4 Mini Homogenizer (Thermo Fisher Scientific, Waltham, MA, USA) following the manufacturer’s protocol, and total RNA was isolated using a Qiagen Rneasy Plus Micro Kit (Qiagen, Hilden, Germany). Genetic DNA was removed using Ambion® DNA-free™ DNase Treatment and Removal Reagents (Thermo Fisher Scientific, Austin, TX, USA) prior to cDNA synthesis with an iScript cDNA synthesis kit (Bio-Rad Laboratories, Hercules, CA, USA). Then, 1 µg of cDNA was used for RT-PCR with SYBR Green FAST Master mix on a Quant Studio 3 real-time PCR Detection System (Applied Biosystems, Waltham, MA, USA). The mRNA expression of pro-inflammatory markers, including inducible nitric oxide synthase (*inos*), interleukin-6 (*il-6*), and interleukin-1 (*il-1*) were assessed. The expression of target genes was normalized to 18S mRNA (Appendix A).

### 2.11. Enzyme-Linked Immunosorbent Assay (ELISA)

Cytokine levels in culture media were determined using an enzyme-linked immunosorbent assay (ELISA) kit (eBioscience, San Diego, CA, USA) following the manufacturer’s instructions. The absorbance was measured at a wavelength of 450 nm using a microplate reader (Molecular Devices, Sillicon Valley, CA, USA). The levels of IL-6 and TNF-α were calculated based on the standard curve and expressed in pg/mL.

### 2.12. Statistical Analysis

All statistical analyses were performed using Prism version 10.4.1 software. All experiments were repeated independently at least three times. Each intervertebral disc yielded two tissue slices. Depending on the assay, three to six disc slices were used per experimental condition. For RT-qPCR, three disc slices were pooled per sample for analysis. In the macrophage co-culture experiments, each experiment was performed three times, with four disc slices per well and three wells per condition. The results are presented as the mean ± standard error of the mean (S.E.M). A *p*-value of less than 0.05 was considered statistically significant. Unpaired *t*-tests were used for comparisons between two groups. For comparisons among three or more groups, one-way ANOVA followed by multiple comparison tests was used. Two-way ANOVA with multiple comparisons was applied when assessing the effects of two independent variables across multiple groups or time points.

## 3. Results

### 3.1. Establishment of a New Disc Slice Culture Model

Here, we developed a new disc slice culture model to enable ex vivo analysis of disc biology and disc degeneration, capturing key features of the in vivo disc microenvironment. To provide physical support to the soft disc tissue, the discs were embedded in 6% agarose prior to sectioning on a vibratome, followed by re-encapsulation with an additional layer of 3% agarose after slicing to prevent NP detachment (Figure 1a). As optimal thickness of a tissue slice is important for maintaining tissue viability and downstream applications [26], we first optimized the thickness of disc slice. Disc slices of 300 μm thickness preserved the AF/NP structural integrity for up to 5 days ex vivo (Appendix A). In contrast, slices thinner than 200 μm failed to maintain a satisfactory AF/NP structure (Appendix A), while slices thicker than 400 μm were difficult to section (Appendix A).

The axial disc slice retained structural features comparable to those of the whole disc, including a centrally located NP and a concentric annulus fibrosus (AF) in the surrounding region (Figure 1b, Appendix A). These representative images further illustrate the advantages of the slicing method: While the whole disc enclosed by the cartilage endplate limits visualization of internal regions, the disc slice reveals distinct AF and NP boundaries (Figure 1b), enabling imaging of spatial–temporal, cell–matrix and cell–cell interactions ex vivo. Moreover, disc slice culture exposes the NP and AF to enable direct detection of cell–cell or cell–tissue interactions that are otherwise inaccessible in whole disc organ culture. For example, this method mimics macrophage infiltration in a spatiotemporally controlled manner (Figure 1c).

### 3.2. Tissue Slice Culture Maintains Disc Structure for up to 5 Days

Bright-field imaging confirmed well-preserved disc structure throughout 5-day culture period, with initial signs of degeneration on day 5 (Figure 2a). Safranin-O/Fast Green staining showed a decreasing trend of proteoglycan content in the AF (orange to red color) (Figure 2b), corroborating histological scoring [8,27,33] on day 5; however, these changes remain statistically insignificant (*p* > 0.05) (Figure 2d). Quantification of the AF area indicated minimal structural changes, with only small clefts and fissures in collagen fibers observed on day 5 (Figure 2e). In addition, PLM imaging of ABPR-stained disc slices revealed similar collagen fiber orientation across various time points (Figure 2c). While PLM does not allow for the identification of specific collagen subtypes, the observed green-to-red birefringence intensity ratio (Figure 2f) indicates that the relative proportion of less mature (green) to more mature (red) collagen fibers remained largely consistent over time. This was further corroborated by Alcian Blue/Picrosirius Red staining (Appendix A), showing the clear NP-AF for up to 5 days, with concentric collagen fibers (red) surrounding the jelly-like NP tissue in the center with abundant proteoglycan (blue).

### 3.3. Disc Slice Culture Maintained Disc Cell Viability for up to Day 5

We next tested the viability of tissue slices over time in comparison to the traditional whole-disc culture format. Using a Live/Dead assay, the percentages of live cells (green) in the NP (90 ± 6.2%) and AF (86 ± 2.7%) regions of disc slices on day 5 were comparable to those in fresh tissue slices (*p* > 0.05) (Figure 3a–c). The results were consistent with those observed in whole disc cultures, as both exhibited no significant changes from their respective baselines (Appendix A). In slices, the percentage of apoptotic cells in both regions increased from day 0 to day 5 (NP: 0.69 ± 0.44% in fresh slices vs. 6.6 ± 2.8% on day 5; AF: 1.0 ± 0.74% in fresh slices vs. 5.5 ± 1.0% on day 5). However, this increase was statistically significant only in the AF region on day 5, with no significant differences observed in the NP across time points (*p* > 0.05) (Appendix A).

LDH activity increased significantly by day 5 in both the disc slice and whole disc culture groups compared to day 0 (*p* < 0.05). The whole disc group showed an increase by day 3, while the disc slice group showed changes by day 5. The LDH levels in culture media were comparable between the disc slice and whole disc groups (*p* > 0.05) (Figure 3d). These results demonstrate that the disc slice culture maintains disc structure and cell viability that are comparable to whole disc culture in ex vivo condition.

### 3.4. Disc Slice Culture Preserves Matrix Proteins

Collagen II and aggrecan are two major ECM proteins in discs, and we interrogated the stability of their mRNA expression over time in the cultures. We also examined the expression of matrix metalloproteinase 13 (*mmp13*), a key catabolic enzyme that degrades collagen II, and released glycosaminoglycans (GAG), essential components of aggrecan, to assess matrix turnover and degradation. Day 0 samples were collected immediately after slicing and served as freshly sliced tissue controls. On day 5, the mRNA expression levels of collagen II (*col2α1*) and aggrecan (*acan*) were increased 7.5 ± 3.3 and 1.0 ± 0.39 folds compared to those in fresh tissues, respectively, but did not reach statistical significance (*p* > 0.05) (Figure 4a,b). Meanwhile, the mRNA expression of *mmp13* remained relatively stable across all time points (*p* > 0.05) (Figure 4c). The cumulative GAG release in culture media significantly increased (~40 fold from baseline) by day 5 in both disc slice and whole disc culture, with no significant differences observed between the two groups (Figure 4d). Therefore, disc tissue slices culture maintained extracellular matrix over the culture period.

### 3.5. Disc Slices Respond to Inflammatory Cytokine IL-1β Stimulation

To investigate the functional responses of disc tissue slices to external stimuli, we treated the slices with the inflammatory cytokine IL-1β and assessed changes in gene expression. IL-1β was chosen for its established role in disc inflammation. In this slice culture model, which is more sensitive than whole disc explants, we used a lower concentration range (1–100 ng/mL) to model early degenerative changes. Higher concentrations resulted in excessive degeneration, limiting interpretability. After tissue slicing, disc slices were cultured in growth media for 1 day and then treated with varying doses of IL-1β, followed by incubation for an additional 24 h (Figure 5a). As shown in Figure 5b, the released GAG in culture media increased in a dose-dependent manner. The mRNA level of *acan* reduced dramatically to 0.27 ± 0.05 at the highest IL-1β dose (Figure 5c), while *col2α1* levels began to decline at IL-1β doses of 100 pg/mL (Figure 5d). The mRNA expression levels of *mmp13*, *il-6*, and *adamts4* increased after IL-1β treatment but did not show statistically significant differences across all doses (*p* > 0.05) (Figure 5e–g). These findings suggest that, upon inflammatory stimulation, tissue slices undergo typical degenerative changes including decreased anabolic and increased catabolic activities.

### 3.6. Disc Slices Activate Macrophages Toward Pro-Inflammatory Phenotype

Previous studies have demonstrated that intervertebral discs can drive macrophages toward a pro-inflammatory M1 phenotype, characterized by distinct morphological and molecular changes [10,18,32]. Similarly, disc slices activated macrophages. Tissue slices were cultured for 3 days and then co-cultured with quiescent murine macrophages for 24 h. As visualized under a bright-field microscope, macrophages exhibited activated morphology after co-culture (Figure 6a), displaying an elongated shape with an increased cytoplasmic spreading compared to the control group (Appendix A). Quantitative analysis revealed significant differences in cell diameter and aspect ratio between the control macrophages and the macrophages + disc slice co-culture group (Figure 6b,c).

Further analysis of gene expression in macrophages demonstrated a significant increase in mRNA expression of pro-inflammatory markers—including *inos* (90 fold, *p* = 0.05), *il-6* (50 fold, *p* = 0.01), and *il-1* (1200 fold, *p* = 0.02)—in the co-culture group compared to the macrophage control group (Figure 6d–f) after a 3-day ex vivo culture. However, mRNA level of *tnf-α* remained unchanged between the two groups (Figure 6g). At the protein level, expression of TNF-α and IL-6 in culture media rose by 4-fold (*p =* 0.05) and 120-fold (*p* = 0.05), respectively, compared to macrophage only group (Figure 6h,i). Released nitric oxide in culture media also increased by ~50 fold (*p* = 6 × 10^−5^) (Figure 6j). These findings indicated that disc slices stimulated macrophages toward inflammatory phenotypes.

### 3.7. Macrophages Infiltrate the Disc Tissue Slices

We and others have reported macrophage infiltration into degenerated disc tissue of humans and mice and its role in aggravating the degenerative process [7,8,10,14,18,34]. To investigate the potential to model this process ex vivo, we performed macrophage overlay experiments (Figure 7a). Disc slices were cultured ex vivo for 3 days before Dil dye-labeled macrophages were overlaid onto the tissue and incubated for 1 day to allow for the disc–macrophage interaction. The numbers and depth of macrophages in tissue slices were assessed in situ under confocal microscopy. As shown in Figure 7, macrophages (red) were observed at depths of up to 96 µm in the AF region and 108 µm in the NP region, respectively. Three-dimensional reconstruction revealed that most macrophages were localized around 50 µm beneath the tissue surface (Figure 7c,d). Quantitative analysis showed that in the NP region—presumably due to its softer and looser matrix—macrophage penetration depth varied widely, reaching up to 110 µm. In contrast, the AF region, which is denser and structurally tighter, exhibited a Gaussian-like distribution of macrophages concentrated around mid-depths (20–99 µm), suggesting restricted but organized infiltration patterns (Figure 7e,f; Appendix A). It is worth noting that the microscope detection limit was approximately 100 µm in depth.

To confirm cell migration at different depths within the disc tissue, we further cryo-sectioned the disc slices into 10 µm thick sections and visualized macrophage distribution across the sections. As shown in Appendix A, macrophages were identified at different depths within the tissue based on serial 10 µm cryosections. In this sample, the penetration was observed up to 50 µm from the superior or inferior boundary, which is consistent with the peak distribution seen in the 3D images. In contrast, when the macrophage overlay experiment was performed on intact discs, no macrophages were detected within the whole disc tissue (Appendix A) but only on the surface. These findings demonstrate that the disc slice culture system enables macrophages to infiltrate both the NP and AF regions, a capability less readily observed in intact whole disc cultures. This model offers a more accessible and physiologically relevant platform for studying macrophage migration and infiltration in the context of disc inflammation and degeneration.

## 4. Discussion

Intervertebral discs function as cushions in the spine, providing both flexibility and load distribution. They are the largest avascular organs, consisting of a gel-like NP at the center, surrounded by concentric collagenous AF layers, with endplates connecting the discs to adjacent vertebrae. Studies have shown that by the age of 50, over 90% of individuals exhibit signs of disc degeneration [35]. This degenerative process often begins early in life, with approximately 37% of asymptomatic individuals showing disc degeneration by the age of 20, and its prevalence increases with each subsequent decade [36]. Despite its clinical significance, effective therapies for disc degeneration remain unavailable, largely due to the lack of suitable models for studying its underlying mechanisms.

Tissue slice culture has been widely used in other fields—such as brain, liver, lymph nodes, cartilage research—to preserve native tissue architecture, maintain cell–cell interactions, and retain the cell–matrix microenvironment. For example, Belanger et al. employed acute lymph node slices to investigate immune responses in a controlled environment [26], and Wang et al. conducted a similar study to evaluate antifibrotic therapies for liver fibrosis and cirrhosis [37]. All these critical features are often lost in traditional two-dimensional (2D) culture systems. Compared to these culture systems, which often fail to replicate the intricate microenvironments of living tissues, tissue slice models offer a more physiologically relevant platform and preserve key organ-specific features including metabolic activity, tissue homeostasis, and certain immunological functions that are essential for cell viability and cell–matrix integrity [26,38,39,40]. Furthermore, ex vivo tissue slice models provide direct access to internal regions of organs, allowing real-time observation of ECM remodeling and microenvironmental changes, critical aspects for studying degenerative processes and evaluating therapeutic interventions [24,41].

While animal models capture the actual biological environment, they often face challenges in isolating specific mechanisms of disc degeneration and tissue repair. We and others have developed disc organ culture systems using bioreactors and organ-on-chip, providing valuable platforms to study disc biology and pathophysiology [27,42,43,44,45]. However, these studies used whole discs, which fail to mimic the disrupted disc structure characteristic of degenerative disc diseases. In this study, we developed a disc tissue slice culture system that captures key features of in vivo conditions by preserving native disc structure, as well as cell–cell and cell–matrix microenvironment (Figure 2 and Figure 3, and Appendix A). When disc slices were cultured ex vivo, tissue retained structural integrity for up to 5 days, with histological staining confirming proteoglycan and collagen retention (Figure 2, Appendix A). Disc cells were viable with minimal apoptotic cells in both the NP and AF regions (Figure 3). Additionally, disc slices maintain similar mRNA levels of *collagen II* and *aggrecan* to the fresh slices, alongside a decrease in matrix degradation enzyme *adamts4* (Figure 4). These data suggest that disc slice culture is a validated model system to study disc biology.

The inflammatory microenvironment plays a critical role in the disease progression. In this microenvironment, disc cells produce increased levels of pro-inflammatory cytokines (TNF-α, IL-1β, IL-6, MCP-1) and MMPs, which recruit inflammatory cells and degrade ECM. These factors collectively exacerbate the progression of disc degeneration and associated pain. IL-1 has been shown to induce the production of catabolic cytokines and enzymes in disc cells and organ cultures [46]. As shown in Figure 5, treatment of disc tissue slices with increasing doses of IL-1β led to a marked reduction (by ~3–4 fold) in *aggrecan* mRNA levels and an increase in released GAG content in the culture media. While *mmp13* and *adamts4* mRNA levels showed an upward trend, the changes were not statistically significant. These findings confirmed the retained metabolic activity of the tissue slices.

TNF-α and IL-6 are well-recognized inflammatory mediators linked to pain severity and inflammation in patients with lower back pain [47]. In our macrophage–disc slice co-culture model, quiescent macrophages exhibited an activated phenotype characterized by elongated cell morphology, increased cell body size, and higher aspect ratios. This activation was accompanied by a significant upregulation of pro-inflammatory cytokines at both the mRNA and protein levels, including IL-6, TNF-α, and nitrite. Notably, while *tnf*-α mRNA expression remained comparable between quiescent and co-cultured macrophages, ELISA revealed a ~4-fold increase in TNF-α protein levels, and IL-6 protein rose by ~120-fold in the culture media (Figure 6), demonstrating that our model successfully recapitulates key features of inflammation relevant to disc-related pain.

A key advantage of the disc slice culture is its ability to study disc-macrophage interaction. Our previously developed disc-organ-on-chip models maintained disc health span up to one month in an in vitro culture, providing a valuable platform to study disc biology and pathology [27]. However, these systems were limited in their ability to investigate the direct crosstalk between inflammatory cells and disc cells. In contrast, the disc slice model enables macrophages to interact directly with both the NP and AF tissues by migrating into the tissue, mimicking in vivo macrophage infiltration observed in degenerative discs. After overlaying the dye-labeled macrophages on top of the disc tissue slices, macrophages migrated towards the central part of the gelatinous NP up to ~100 µm deep, which is the maximum detection depth of our confocal laser system. Macrophages were also detected in the AF region, by migrating through the gaps between the collagen fibers. A greater number of macrophages would likely be detected at deeper depths if the culture period was extended beyond 1 day, as the majority were accumulated within the top 50 μm (Figure 7). This finding suggests that disc slices serve as a great tool for investigating the role of macrophages in the disease course of disc degeneration, allowing for real-time and spatiotemporal analysis of macrophage infiltration as well as therapeutic screening.

While our disc slice culture system provides a valuable platform for studying pathological progression and disc–macrophage interactions, several limitations should be considered. Beyond macrophage infiltration, other pathological processes—such as nerve ingrowth and angiogenesis—also contribute to disc degeneration, and it remains to be tested how best to model these ex vivo systems. Since disc degeneration is often observed in both symptomatic and asymptomatic individuals [15,35], the relationship between immune responses and disc pathology is complex and remains an active research area in our laboratory. Our short-term culture model captures early disc responses, but long-term studies are needed to fully understand disc degeneration. Future work extending the culture period to 7–14 days may further clarify chronic degenerative changes and better establish the model’s relevance for long-term disc pathology. To address this, we are developing a microfluidic disc-on-a-chip model that will allow continuous nutrient delivery, precise environmental control, and investigations of key degenerative processes, such as neovascularization. This system will provide deeper insights into disc pathology and potential therapeutic interventions. In addition, while mechanical loading is known to be essential for intervertebral disc homeostasis and degeneration, our current tissue slice model prioritizes preserving matrix structure and cellular viability in a simplified setting. Incorporating mechanical stimulation into such a system may require substantial engineering adjustments due to the fragility of tissue slices. Future development may build on our prior Disc-on-a-Chip^MF^ platform [42] to adapt dynamic loading for tissue slices. While this ex vivo model does not capture behavioral pain outcomes, it provides a controlled system to study disc-intrinsic inflammatory and degenerative changes that are mechanistically relevant to pain development in vivo. Future experiments will include pain-related measures and functional assessments, including nerve ingrowth, cytokine production, and neuropeptides release. Lastly, tissue swelling may affect matrix integrity and cellular responses in our culture system, although we did not observe a dramatic change. Future optimizations of culture conditions and biomechanical constraints are necessary to minimize swelling artifacts and replicate the disc microenvironment. Addressing these limitations will enhance the translational potential of our disc slice model for studying disc degeneration and disease mechanisms.

## 5. Conclusions

In summary, we developed a novel disc slice culture system that preserves key structural and biological features of the intervertebral discs while allowing controlled experimentation. This ex vivo model successfully maintained tissue integrity, cell viability, and ECM composition over a 5-day culture. A significant advantage of this system is its ability to facilitate macrophage infiltration, a process that is otherwise not feasible in intact whole-disc cultures. Additionally, the model is well-suited for studying inflammatory responses and degeneration mechanisms, offering a controlled yet physiologically relevant alternative to conventional models.

## Figures and Tables

**Figure 1 cells-14-01230-f001:**
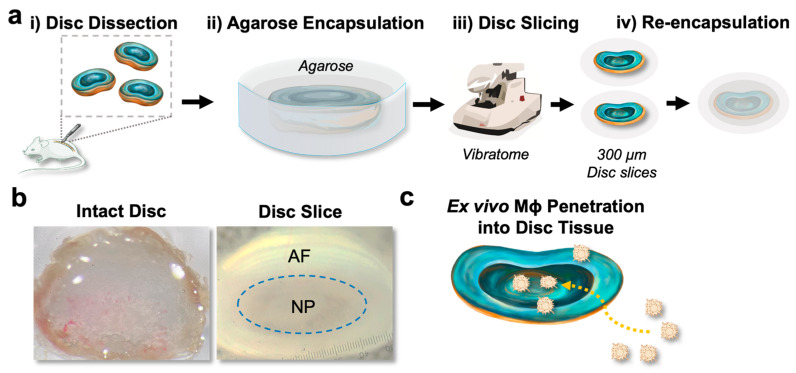
Disc tissue slicing offers a versatile platform for downstream applications. (**a**) Shown here is a schematic representation of the disc tissue slicing process, which includes the follow steps: (i) dissecting intact intervertebral discs from C57BL/6J mice aged 10–12 weeks; (ii) encapsulating disc tissues in 6% agarose gel on ice; (iii) sectioning to 300 µm disc slices with a vibratome; and (iv) re-encapsulating with 3% sucrose to seal the NP. (**b**) Representative microscopic images of intact and disc tissue slice are depicted. The whole disc, enclosed by the intact cartilage endplate, limits direct visualization of the annulus fibrosus (AF) and nucleus pulposus (NP). In contrast, the disc slice reveals distinct and well-defined AF and NP boundaries, enabling direct visualization and analysis. (**c**) A cartoon illustrates macrophage (yellow) penetration into the disc slice (blue), suggesting functional applications of the disc tissue slice and its potential for studying cell–cell and cell–matrix interactions in a spatial temporal manner. Scale bar: 200 µm.

**Figure 2 cells-14-01230-f002:**
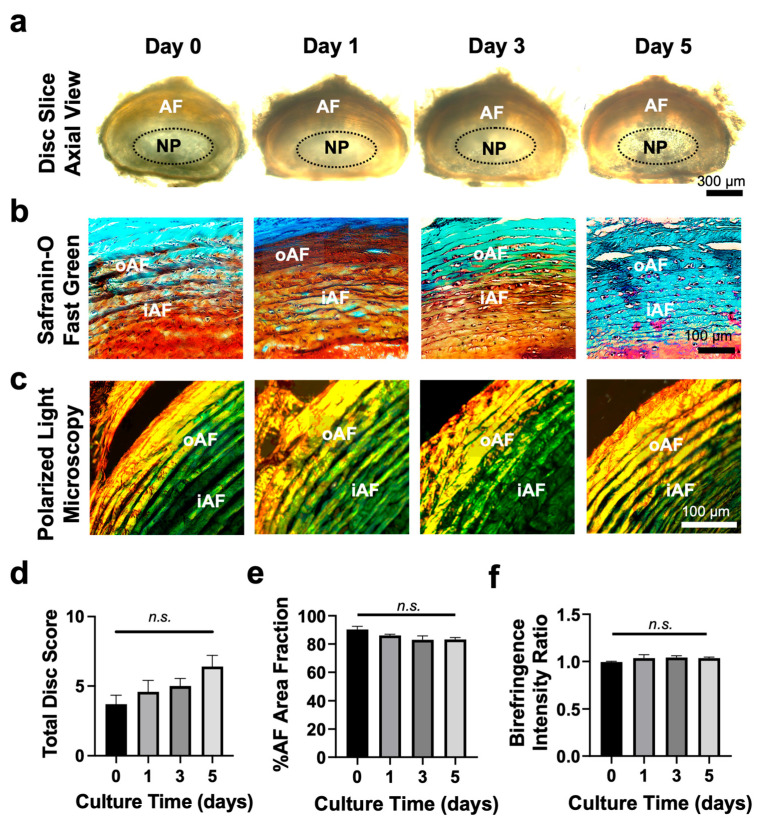
Ex vivo disc slice culture maintains disc matrix integrity for up to five days. (**a**) Representative bright-field microscopic images showed that disc slices (~300 μm thick) maintained concentric collagen fibers surrounding the soft NP tissues in a 48-well plate for up to 5 days. (**b**) Representative Safranin-O/Fast Green histological images showed no obvious tissue damage in both outer AF (oAF) and inner AF (iAF) regions for up to 5 days. (**c**) Representative polarized light microscopy (PLM) images revealed well-organized collagen fibers in AF with similar structural organization and birefringence properties throughout the culture period. (**d**) Histopathological scoring of disc slices showed a slight increasing trend from day 0 to day 5, with no statistically significant difference between any two time points. (**e**) Quantitative analysis of the AF area fraction revealed minimal changes from day 0 to day 5. (**f**) The PLM images demonstrated a similar collagen type as indicated by green-to-red color ratio at various time points, indicating comparable collagen composition and maturity levels, with green representing less mature collagen and red indicating more mature collagen. Each experiment was repeated at least three times. In each experiment, six disc slices were used per time point. n.s., not significant by ordinary one-way ANOVA with multiple comparisons. Data represent mean ± SEM.

**Figure 3 cells-14-01230-f003:**
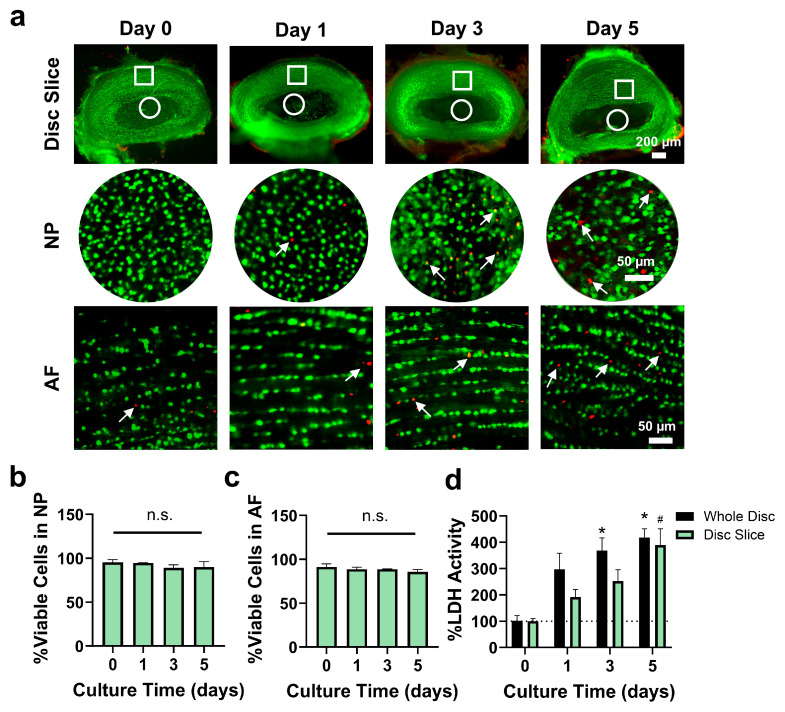
Disc cells are viable up to 5 days of disc slice culture. (**a**) Cell viability was assessed with a Live/Dead assay Kit (*n* = 3 per time point). Green and red represent live and dead cells, respectively. Arrows indicate dead cells (red). (**b**,**c**) Percentages of viable cells showed that cell viability remained stable over time for up to 5 days of ex vivo culture in AF and NP. (**d**) Normalized LDH activity was measured in the disc slices and whole disc culture media for up to 5 days. Whole disc culture exhibited significantly higher LDH activity from day 3, while disc slice culture showed increased LDH activity on day 5, when compared with their respective day 0 controls. Data represent mean ± SEM. One-way ANOVA with multiple comparisons was performed in (**b**,**c**), and two-way ANOVA with multiple comparisons was performed in (**d**). Four disc slices were used for each time point. Experiments were repeated at least three times. n.s., not significant; * *p* < 0.05 vs. day 0 in whole disc group; ^#^ *p* < 0.05 vs. day 0 in disc slice group.

**Figure 4 cells-14-01230-f004:**
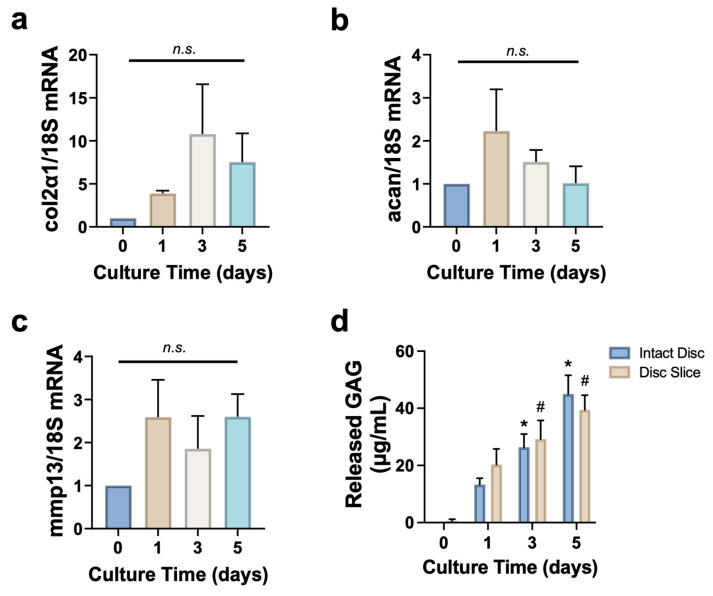
Disc slice culture preserves the expression of matrix genes. (**a**,**b**) mRNA expressions of anabolic genes *collagen II* (**a**), *aggrecan* (**b**), and catabolic gene *mmp13* (**c**) were comparable among various times points for up to day 5. The expression of target genes was normalized to 18S. (**d**) The cumulative released GAG was increased in culture media with comparable increasing levels between whole disc and disc slice culture for up to 5 days. Data represent mean ± SEM. One-way ANOVA with multiple comparisons was performed in (**a**–**c**) and two-way ANOVA with multiple comparisons was performed in (**d**). Each experiment was repeated at least three times. In each experiment, twelve disc slices were cultured per time point with four disc slices per well and three wells in each time point. n.s., not significant; * *p* < 0.05 vs. day 0 in whole disc group; ^#^ *p* < 0.05 vs. day 0 in disc slice group.

**Figure 5 cells-14-01230-f005:**
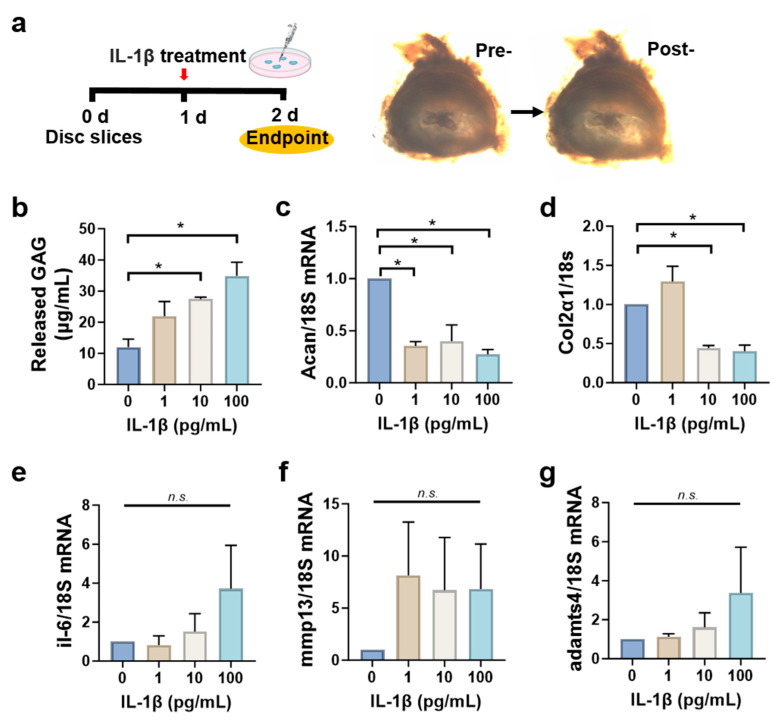
Disc slices respond to IL-1β treatment. (**a**) A schema for experimental design is shown alongside representative images of disc slices before and after 24 h treatment with IL-1β. (**b**) Released GAG in culture media showed a dose-dependent increase in catabolic activity in response to IL-1β stimulation. The mRNA levels of *aggrecan* (**c**) and *col2α1* (**d**) declined significantly in response to IL-1β treatment, while the mRNA levels of *il-6* (**e**), *mmp13* (**f**), and *adamts4* (**g**) did not show statistically significant difference among various IL-1β dosages, despite an increasing trend. Note: Data represent mean ± SEM. Each experiment was repeated at least three times. For each treatment dose, sixteen disc slices were used with four disc slices per well and four wells for each dose. n.s., not significant. * *p* < 0.05 compared to fresh tissue slices (day 0) via one-way ANOVA. The expression of target genes was normalized to 18S.

**Figure 6 cells-14-01230-f006:**
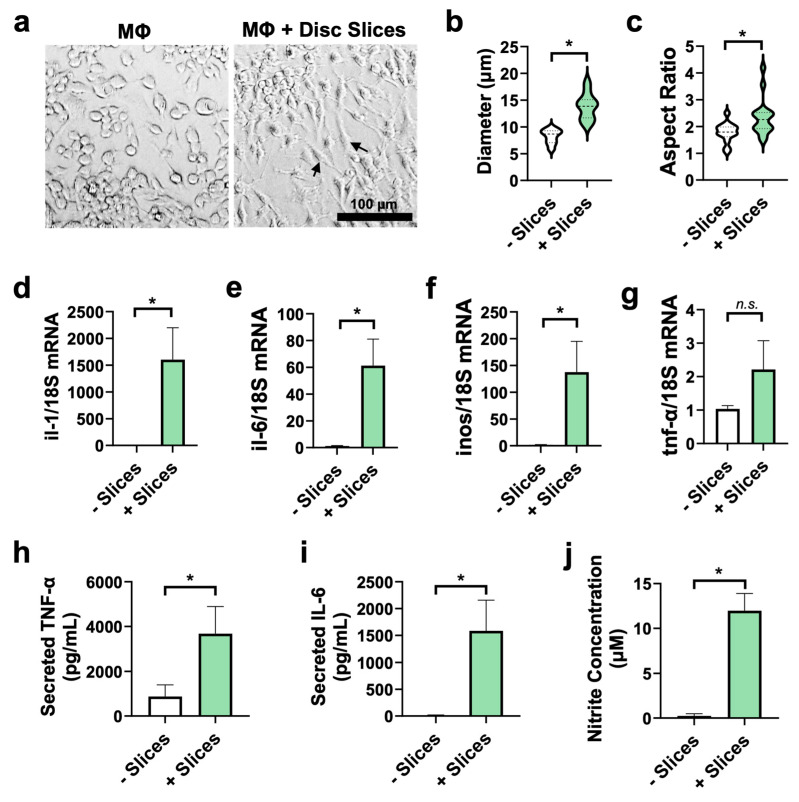
Disc slices promote macrophages toward pro-inflammatory phenotype in a co-culture environment. (**a**) Representative images of macrophages before and after co-culture with disc slices for 1 day. The cell bodies of activated macrophages were irregular and stretched compared to the quiescent macrophages. (**b**,**c**) Quantification analysis of macrophage cell diameter and aspect ratio showed larger cell body and more elongated and activated morphology after coculture with disc slices. (**d**–**g**) The mRNA levels of *il-1*, *il-6*, and *inos* were significantly increased in the macrophages co-cultured with disc slice compared to the control macrophages. The expression of *tnf-α* shows an increased trend but did not reach statistical significance. (**h**,**i**) Secreted TNF-α and IL-6 cytokines increased significantly in the culture media of co-culture group compared to the control detected by ELISA. (**j**) Nitrite levels in the media were significantly higher in the co-culture group compared to the macrophage only group. Note: *n* = three experiments in (**d**–**j**); 8–15 ROIs pooled from three experiments in (**b**,**c**). Data represent mean ± SEM. Each experiment was repeated at least three times. In each experiment, three wells of pre-seeded macrophages were used to co-culture with or without disc slices (four disc slices per well). * *p* < 0.05; n.s., not significant via unpaired t-test.

**Figure 7 cells-14-01230-f007:**
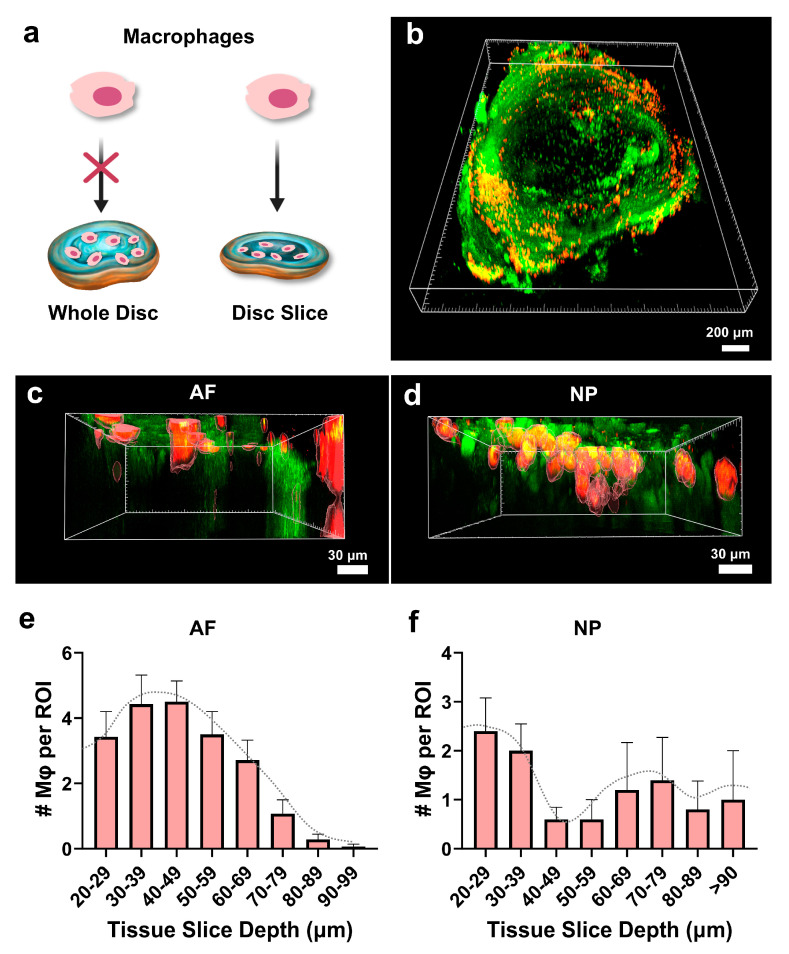
Disc slices allow for direct visualization of macrophage infiltration. (**a**) Schema of experimental design is illustrated. (**b**) Three-dimensional (3D) images depict macrophage distribution in overlay experiments. Macrophage infiltration (DiI dye labeled Raw 264.7 macrophages, red) was evident in the NP region, showing their dispersion across the loose NP structure. (**c**) Three-dimensional reconstruction of AF region depicts the infiltration of macrophages into disc slices. Clusters of cells were observed at various depth (perpendicular to disc axial surface) into the slices. (**d**) Three-dimensional reconstruction of NP region shows macrophage infiltration into different depths of NP structure. (**e**,**f**) Quantification of the number of macrophages at various depths of AF and NP regions is presented here, up to a depth of 100 µm. Note that 100 µm is the detection limit of the microscope. Each experiment was repeated at least three times with at least three disc slices in each experiment. Scale bars: 30 μm.

## Data Availability

Data supporting the findings of this study are available upon request.

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
