# Peer review of "An Ex Vivo Intervertebral Disc Slice Culture Model for Studying Disc Degeneration and Immune Cell Interactions"

_cells, 2025, doi:10.3390/cells14161230_

Round 1

Reviewer 1 Report

Comments and Suggestions for Authors

The manuscript reports an organotypic culture model of mouse IVDs. The model is comprehensively characterized and is demonstrated in the application in studying macrophage infiltration. The manuscript is well-presented and is innovative. There are several comments the authors may consider as suggestions.

The slice is embedded in agarose for culture (line 114~117). Does agarose influence the IL-1β permeation? Also, when used for studying macrophage infiltration, macrophage may infiltrate the agarose instead of IVD matrix.

In the conclusion, it is claimed that macrophage infiltration is “a process that is otherwise not feasible in intact whole disc culture”. This may not be true because stem cell migration into the IVD has been investigated using whole organ culture. (PMID: 31002939) Also the naïve boundary of IVD is already damaged by vibratome sectioning, the macrophage infiltration may be “exaggerated” or artificially amplified compared to in vivo situation.

Other comments

In the rt-qPCR essay of macrophages, how macrophages were separated from the IVD slice for RNA isolation? How to avoid bias if some “infiltrating” macrophages were not collected?

What is the section thickness for PLM imaging? In the image analysis, how the ROIs were defined?

The “n” values are missing for the result and statistic. How many biological replicates (mice)? And how many technical replicates?

In figure 5, some are IL-1β and some panels have x axis title as IL-1 dose. Are they different?

Author Response

The manuscript reports an organotypic culture model of mouse IVDs. The model is comprehensively characterized and is demonstrated in the application in studying macrophage infiltration. The manuscript is well-presented and is innovative. There are several comments the authors may consider as suggestions.

Comment 1.1: The slice is embedded in agarose for culture (line 114~117). Does agarose influence the IL-1β permeation? Also, when used for studying macrophage infiltration, macrophage may infiltrate the agarose instead of IVD matrix.

Response 1.1: We appreciate the reviewer raising this important point.

  1. IL-1β Stimulation Experiments: To maintain tissue integrity and prevent dehydration during the culture period, a drop of 2% agarose was applied on top of the tissue slices in the IL-1β stimulation experiments. We acknowledge that agarose hydrogel concentration can influence molecular diffusion; higher concentrations result in reduced infusibility due to a tighter polymer network and smaller mesh size [Liang et al., Journal of Controlled Release, 2006; 115(2):189–196]. However, the mature form of IL-1β (~17 kDa) is relatively small and has been shown to diffuse through agarose hydrogels without significant restriction [Pluen et al., Biophysical Journal, 1999; 77(1):542–552]. Thus, we believe that the presence of agarose did not substantially hinder IL-1β diffusion into the disc tissue.
  2. Macrophage Overlay Experiments: We’d like to clarify that the macrophage overlay experiments were performed without applying agarose on top of the tissue slices. The slices were embedded only around the sides in agarose to provide support and structural stability vibratome sectioning, but the top surface remained fully exposed. This design ensured direct contact between the macrophages and the tissue surface, eliminating concerns about macrophage infiltration into agarose rather than the tissue itself. To further minimize nonspecific adhesion, we rinsed the slices thoroughly with PBS three times after incubation to remove any floating or loosely attached cells. 3D Z-stack imaging was initiated from the exposed disc surface to verify that macrophage signals originated within the tissue. Additionally, cryo-sectioning of disc slice after the co-culture confirmed macrophage infiltration into the NP and AF regions, supporting that migration occurred into the intervertebral disc matrix, not the surrounding agarose.

We have added this clarification to the “Additional Discussion on Macrophage Overlay” section on page 11 of the Supplemental Materials:Regarding macrophage infiltration, agarose was not applied on top of the tissue slices in the macrophage overlay experiment. Instead, the disc slices were only stabilized around the sides with agarose, leaving the top surface fully exposed for direct contact with macrophages. Therefore, the experiment did not involve agarose encapsulation that would restrict access or alter diffusion. While it is theoretically possible that some macrophages could migrate into the lateral agarose boundary, this is unlikely given the observed infiltration pattern and the experimental design. To further reduce the possibility of non-specific surface attachment, the slices were thoroughly rinsed three times with a large volume of PBS to remove any loosely attached or floating cells prior to imaging. During 3D imaging, the disc surface was used as the starting point for Z-stack acquisition to ensure macrophage localization was assessed within the tissue. Additionally, cryo-sectioning and imaging confirmed that macrophages had penetrated the nucleus pulposus (NP) and annulus fibrosus (AF) regions. Taken together, these steps demonstrate that macrophages infiltrated the intervertebral disc tissue itself, and that the surrounding agarose scaffold had no significant impact on this process.”

We also added “To facilitate direct macrophage interaction, tissue slices used in the overlay experiments were not capped with agarose on the top surface.” in the Methods 2.8 section on page 5, to clarify this difference in experimental setup.

Comment 1.2: In the conclusion, it is claimed that macrophage infiltration is “a process that is otherwise not feasible in intact whole disc culture”. This may not be true because stem cell migration into the IVD has been investigated using whole organ culture. (PMID: 31002939) Also the naïve boundary of IVD is already damaged by vibratome sectioning, the macrophage infiltration may be “exaggerated” or artificially amplified compared to in vivo situation.

Response 1.2: We appreciate the reviewer’s thoughtful comment and the reference to studies utilizing whole organ culture models. Intact, healthy discs with preserved annulus fibrosus and nucleus pulposus structures typically pose a substantial barrier to immune cell infiltration due to their dense extracellular matrix. In the cited study, disc degeneration was induced by mechanical loading, which may disrupt the disc matrix and may facilitate cell infiltration.  Similarly, when disc herniates, the AF/NP structure is disrupted and macrophages can infiltrate into disc tissues.

We acknowledge that vibratome-sectioned tissue creates a disrupted boundary and expose the disc tissue to environmental cues, which is one of the objectives of this study. To avoid overstating our conclusions, we have revised the discussion to clarify that the observed macrophage infiltration reflects a permissive, yet still structurally relevant, model of disc injury rather than a fully intact disc. This platform allows us to study immune cell-tissue interactions under controlled conditions that approximate the complex in vivo environment. Therefore, we revised the manuscript throughout to avoid the “exaggerated” points. For example, we changed the line 481-483, page 15 in the manuscript to “These findings demonstrate that the disc slice culture system enables macrophages to infiltrate both the NP and AF regions, a capability less readily observed in intact whole disc cultures.”

Other comments

Comment 1.3: In the rt-qPCR essay of macrophages, how macrophages were separated from the IVD slice for RNA isolation? How to avoid bias if some “infiltrating” macrophages were not collected?

Response 1.3: We appreciate the reviewer’s insightful question. We fully agree with this reviewer that we may not collect all macrophages, especially those macrophages infiltrated into the disc slices. However, we tried our best to collect all macrophages in the culture well and surface of tissue slices. In our future experiments, we can label macrophages and sort cells from tissue slices for further analysis.  To clarify this procedure, we have added the following description to the Supplemental Materials (page 1) under the new section titled “Macrophage RNA Isolation Protocol from Co-Culture System”: “Following the experimental incubation period, IVD slices were carefully removed using sterile forceps and rinsed well in Eppendorf tubes for flash freezing. The wells were then incubated in TRIzol for 5 minutes at room temperature to collect all cell lysate. After incubation, the TRIzol solution from each well was collected, and any residual agarose gel was removed via brief centrifugation. Microscopic inspection at the end of the co-culture period showed that the majority of macrophages were adherent to the bottom of the well, with minimal presence on the surface of the IVD slices, and negligible infiltration into the agarose matrix. This suggests that the TRIzol incubation was sufficient to recover the majority of macrophage-derived RNA, including any macrophages associated with the tissue surface. To further control for variability in RNA yield between samples, total RNA concentrations were quantified using a Nanodrop spectrophotometer. Equal amounts of RNA (1 µg/sample) were used for cDNA synthesis and subsequent RT-qPCR analysis.” This approach, while not perfectly isolating every individual macrophage, provides a consistent and reproducible method that captures the predominant population of interest and minimizes sampling bias. We have included these details in the revised Supplemental Materials as suggested and thank the reviewer again for raising this important point.

Comment 1.4: What is the section thickness for PLM imaging? In the image analysis, how the ROIs were defined?

Response 1.4: We appreciate the question. Section Thickness was 10 µm. ROIs were defined as the central AF region. For quantification purposes, the ROIs encompassed the entire field of view captured at ×200 magnification. These information was included on section 2.5, line 152-156 in the manuscript: “Disc tissues were sectioned at a thickness of 10 μm. Images were acquired at ×200 magnification at consistent exposure settings across all samples to ensure comparability. Regions of interest (ROIs) were defined as the central AF region. For quantitative analysis, the entire field of view at ×200 magnification was used as the ROI to ensure consistency across samples.”

Comment 1.5: The “n” values are missing for the result and statistic. How many biological replicates (mice)? And how many technical replicates?

Response 1.5: We appreciate the question. We made sure this information is included in line 234-239 on page 6 and figure legends. The following was added to the Methods section: “All experiments were repeated independently for at least three times. Each intervertebral disc yielded two tissue slices. Depending on the assay, three to six disc slices were used per experimental condition. For RT-qPCR, three disc slices were pooled per sample for analysis. In the macrophage co-culture experiments, each experiment was performed three times, with four disc slices per well and three wells per condition.”

Comment 1.6: In figure 5, some are IL-1β and some panels have x axis title as IL-1 dose. Are they different?

Response 1.6:  Thank you for point this out this mistake. They are the same. We changed all x axis title to IL-1β (pg/mL) for consistency in Figure 5 on page 11.

Reviewer 2 Report

Comments and Suggestions for Authors

Consider extending the culture period beyond Day 5 (e.g., to Day 7–14) to evaluate the model’s applicability for long-term disc degeneration studies.

Author Response

Comment 2.1: Consider extending the culture period beyond Day 5 (e.g., to Day 7–14) to evaluate the model’s applicability for long-term disc degeneration studies.

Response 2.1: We appreciate this great suggestion.  We agree that extending the culture period to 7–14 days would provide valuable insight into the progression and chronicity of disc degeneration and inflammation. We are optimizing our culture condition for a long-term culture. In the current study, we selected a 5-day culture period to specifically capture early degenerative and inflammatory events within a controlled ex vivo environment, during which tissue viability, matrix architecture, and cell morphology remained well preserved. We have added this point to the Discussion section on page 16, line 569-572 in the manuscript: “Our short-term culture model captures early disc responses, but long-term studies are needed to fully understand disc degeneration. Future work extending the culture period to 7–14 days may further clarify chronic degenerative changes and better establish the model’s relevance for long-term disc pathology.”

Reviewer 3 Report

Comments and Suggestions for Authors

Oh et al. present a study utilising ex vivo intervertebral disc (IVD) slice cultures from mice to investigate degenerative changes. This model offers several advantages, including preservation of native extracellular matrix architecture and cellular heterogeneity, and allows for mechanistic exploration under controlled ex vivo conditions. Notably, the authors also demonstrate direct interaction between disc tissue and macrophages in co-culture, providing insight into the role of inflammation in driving IVD degeneration. This adds a valuable dimension to the model by incorporating direct interactions of immunological components often absent in in vitro systems of IVD  degeneration. While the approach is innovative and increasingly relevant in disc research, several important concerns should be addressed:

  1. Given the load-responsive nature of IVD cells (Lazaro-Pacheco et al.,2023), the lack of physiological mechanical stimulation is a notable limitation. Mechanical cues are critical for disc homeostasis and degeneration; therefore, incorporating bioreactor-based dynamic loading would enhance the physiological relevance of the model.
  2. It is unclear whether the authors used healthy, aged, or structurally compromised discs as the starting material. This distinction is essential, as it appears the study primarily demonstrates the initiation of degenerative changes by inflammation, rather than capturing the full spectrum of its progression. Clarifying the baseline condition of the tissue would help contextualise the findings and better define the stage of degeneration being modelled.
  3. While the authors imply that their model recapitulates inflammation-associated IVD degeneration relevant to pain, no functional or behavioural assessments were performed to link the inflammatory responses to pain-related outcomes directly. Without such evidence, the model’s relevance to pain remains speculative. 
  4. Although the inclusion of macrophages offers a vital advancement, the immune component in this model is limited to direct co-culture. It does not reflect the full complexity of immune cell recruitment, trafficking, and resolution observed in vivo. Further, the activation state and source of macrophages and their ratio to disc cells can significantly influence the inflammatory outcome and should be carefully controlled or justified.
  5. The authors utilised IL-1β to induce inflammation; however, it is unclear why IL-1β was selected as the primary inflammatory stimulus in this context. While IL-1β is commonly used in in vitro disc studies, its role as a principal driver of inflammation in IVD degeneration, particularly in early versus late stages, remains a topic of debate. Provide evidence regarding the physiological relevance of the concentration and what endogenous cell types are presumed to produce IL-1β in vivo within the disc microenvironment. Addressing these points would help validate the appropriateness of the inflammatory model employed.
  6. While the authors demonstrate that the macrophages exhibit a pro-inflammatory phenotype, the study does not fully explore the functional consequences of this activation within the tissue context. In vivo, migrating macrophages contribute to the progression of IVD degeneration through the secretion of pro-inflammatory cytokines and mechanisms such as phagocytic activity, matrix degradation, and sustained activation of resident disc cells. It remains unclear whether the macrophages in this model actively engage in such processes, particularly phagocytosis or extracellular matrix remodelling, which are critical to driving structural degeneration. Further demonstration of macrophage behaviour within the slice culture, beyond cytokine expression and macrophage migration, would strengthen the mechanistic interpretation and help clarify whether the observed inflammation reflects early degenerative initiation, tissue damage amplification, or potentially unresolving inflammation.

Author Response

Oh et al. present a study utilising ex vivo intervertebral disc (IVD) slice cultures from mice to investigate degenerative changes. This model offers several advantages, including preservation of native extracellular matrix architecture and cellular heterogeneity, and allows for mechanistic exploration under controlled ex vivo conditions. Notably, the authors also demonstrate direct interaction between disc tissue and macrophages in co-culture, providing insight into the role of inflammation in driving IVD degeneration. This adds a valuable dimension to the model by incorporating direct interactions of immunological components often absent in in vitro systems of IVD  degeneration. While the approach is innovative and increasingly relevant in disc research, several important concerns should be addressed:

Comment 3.1: Given the load-responsive nature of IVD cells (Lazaro-Pacheco et al.,2023), the lack of physiological mechanical stimulation is a notable limitation. Mechanical cues are critical for disc homeostasis and degeneration; therefore, incorporating bioreactor-based dynamic loading would enhance the physiological relevance of the model.

Response 3.1: We appreciate the reviewer’s insightful comment regarding the importance of physiological mechanical stimulation in intervertebral disc (IVD) models. Indeed, mechanical cues are essential for disc homeostasis and degeneration, and we fully acknowledge that the lack of dynamic mechanical loading represents a limitation of the current tissue slice culture model. We will incorporate mechanical loading in our future experiments using our established disc-on-chip model, which include both flow and mechanical loading components (Xie et al., Adv Mater Technol. 2023). These thin discs slices are inherently more delicate and may not tolerate standard compressive loading protocols without deformation or damage. We are in the process of device design that can implement this essential physiological dynamic loading for the ex vivo tissue culture. We have added this point to the revised Discussion section on page 17, line 576-580 of the manuscript:While mechanical loading is known to be essential for IVD homeostasis and degeneration, our current tissue slice model prioritizes preserving matrix structure and cellular viability in a simplified setting. Incorporating mechanical stimulation into such a system may require substantial engineering adjustments due to the fragility of tissue slices. Future development may build on our prior Disc-on-a-ChipMF platform [Xie et al., Adv Mater Technol. 2023] to adapt dynamic loading for tissue slices.”

Comment 3.2: It is unclear whether the authors used healthy, aged, or structurally compromised discs as the starting material. This distinction is essential, as it appears the study primarily demonstrates the initiation of degenerative changes by inflammation, rather than capturing the full spectrum of its progression. Clarifying the baseline condition of the tissue would help contextualise the findings and better define the stage of degeneration being modelled.

Response 3.2:  Thank you for this important comment. The intervertebral discs used in this study were harvested from young (10-16 week-old, as described in section 2.1), healthy mice with no prior structural compromise or degenerative features. The goal of this project is to establish an in vitro culture system to model the early pathological phase of disc degeneration under controlled conditions. We agree that using aged or already compromised discs could represent later stages of disease and offer complementary insights. Future studies may explore extended culture durations and alternative baseline conditions to capture a broader spectrum of degenerative progression. We have clarified this point in the revised manuscript on page 3, line 99-101 of revised manuscript:Discs were isolated from young, healthy mice to establish a baseline of structural and cellular integrity, allowing for the controlled induction and study of early degenerative events.”

Comment 3.3: While the authors imply that their model recapitulates inflammation-associated IVD degeneration relevant to pain, no functional or behavioral assessments were performed to link the inflammatory responses to pain-related outcomes directly. Without such evidence, the model’s relevance to pain remains speculative. 

Response 3.3: Thank you for this thoughtful comment. The goal of our current ex vivo model is to recapitulate inflammation-associated molecular and cellular events in a controlled environment, enabling mechanistic insight into early degenerative processes without systemic variables. Our future experiment will include pain-related measures and functional assessments, including nerve ingrowth, cytokine production, and neuropeptides release. To avoid confusion, we have clarified this scope in the revised discussion on page 17, line 581-586 of revised manuscript: “While this ex vivo model does not capture behavioral pain outcomes, it provides a controlled system to study disc-intrinsic inflammatory and degenerative changes that are mechanistically relevant to pain development in vivo. Future experiment will include pain-related measures and functional assessments, including nerve ingrowth, cytokine production, and neuropeptides release.”

Comment 3.4: Although the inclusion of macrophages offers a vital advancement, the immune component in this model is limited to direct co-culture. It does not reflect the full complexity of immune cell recruitment, trafficking, and resolution observed in vivo. Further, the activation state and source of macrophages and their ratio to disc cells can significantly influence the inflammatory outcome and should be carefully controlled or justified.

Response 3.4: We appreciate these important suggestions.  We agree that while our model represents an important step forward by incorporating macrophages into an ex vivo disc culture, it does not fully replicate the complexity of immune cell recruitment, trafficking, and resolution seen in vivo. Based on the current work, our lab is developing a more advanced ex vivo model that integrates controlled immune cell recruitment, etc to better mimic physiological immune responses within the disc microenvironment, which will be reported in future. We also agree that the activation state, origin, and ratio of macrophages to disc cells can significantly influence outcomes. In the current study, we used the RAW 264.7 murine macrophage cell line, which offers a reproducible and widely accepted model for studying macrophage-mediated inflammatory responses in vitro. In future studies, we plan to incorporate primary macrophages from bone marrow or tissue-resident sources, examine polarized M1/M2 states, and systematically vary cell ratios to better mimic in vivo heterogeneity.

To reflect these limitations and future directions, we have added the following paragraph to the on page 11 as “Additional Discussion on Limitations” of Supplemental Materials: “For macrophage overlay study, while the current model uses Raw 264.7 macrophages in direct co-culture to study inflammation-induced disc degeneration, it does not capture immune cell trafficking or resolution mechanisms. Additionally, the macrophage source, activation state, and cell ratio can influence inflammatory outcomes. Future efforts will involve primary macrophage populations, polarization studies, and more complex platforms that incorporate immune recruitment dynamics to better approximate in vivo conditions.”

Comment 3.5: The authors utilised IL-1β to induce inflammation; however, it is unclear why IL-1β was selected as the primary inflammatory stimulus in this context. While IL-1β is commonly used in in vitro disc studies, its role as a principal driver of inflammation in IVD degeneration, particularly in early versus late stages, remains a topic of debate. Provide evidence regarding the physiological relevance of the concentration and what endogenous cell types are presumed to produce IL-1β in vivo within the disc microenvironment. Addressing these points would help validate the appropriateness of the inflammatory model employed.

Response 3.5: We appreciate the reviewer’s insightful comment. IL-1β was selected as the inflammatory stimulus due to its well-established role in initiating catabolic and inflammatory pathways in intervertebral disc degeneration. Multiple studies, including Le Maitre et al. (Spine, 2005, PMID: 15987475), have demonstrated that IL-1β promotes matrix degradation and pro-inflammatory signaling in disc cells, making it a widely used cytokine in disc degeneration models. In vivo, IL-1β is produced by both resident disc cells (nucleus pulposus and annulus fibrosus) and infiltrating immune cells, particularly macrophages, in response to injury or stress (Risbud & Shapiro, Nat Rev Rheumatol, 2014). These cell populations contribute to the local inflammatory milieu during both early and progressive stages of degeneration. Given that our model utilizes tissue slices rather than intact whole discs, which are more sensitive to exogenous stimuli due to increased surface exposure and diffusion., we employed a lower IL-1β concentration range of 1–100 ng/mL. In preliminary testing, higher concentrations (e.g., 1000 ng/mL) led to excessive degeneration and tissue breakdown without clear dose-dependent effects. The selected range allowed us to induce measurable and graded inflammatory responses while preserving structural integrity. We have added the following clarification to the revised manuscript in section 3.5 on page 11, line 378-381: “IL-1β was chosen for its established role in disc inflammation. In this slice culture model, which is more sensitive than whole disc explants, we used a lower concentration range (1–100 ng/mL) to model early degenerative changes. Higher concentrations resulted in excessive degeneration, limiting interpretability.”

Comment 3.6: While the authors demonstrate that the macrophages exhibit a pro-inflammatory phenotype, the study does not fully explore the functional consequences of this activation within the tissue context. In vivo, migrating macrophages contribute to the progression of IVD degeneration through the secretion of pro-inflammatory cytokines and mechanisms such as phagocytic activity, matrix degradation, and sustained activation of resident disc cells. It remains unclear whether the macrophages in this model actively engage in such processes, particularly phagocytosis or extracellular matrix remodeling, which are critical to driving structural degeneration. Further demonstration of macrophage behaviour within the slice culture, beyond cytokine expression and macrophage migration, would strengthen the mechanistic interpretation and help clarify whether the observed inflammation reflects early degenerative initiation, tissue damage amplification, or potentially unresolving inflammation.

Response 3.6:  We thank the reviewer for this thoughtful and constructive comment. We fully agree that macrophages contribute to intervertebral disc degeneration not only through cytokine secretion but also via functions such as phagocytosis, matrix remodeling, and sustained activation of resident disc cells. We will simulate all these functions in this ex vivo model, the ultimate goal of this project. In our current study, we focused on the establishment of the platform to mimic inflammatory activation and spatial migration of macrophages. However, we acknowledge that further exploration of macrophage behavior would significantly enhance the mechanistic understanding of their role in degeneration. We plan to address this gap in future work within the ex vivo tissue context. To reflect this limitation and future direction, we have added the following as “Additional Discussion on Limitations” in Supplemental Materials on page 11: “While our study demonstrates pro-inflammatory macrophage activation and migration, it does not yet assess downstream functional consequences such as phagocytic activity or matrix remodeling. Future experiments will include these assessments to better delineate the role of macrophages in early versus progressive stages of disc degeneration.”

Round 2

Reviewer 1 Report

Comments and Suggestions for Authors

My questions are well-addressed. I have no more comments.

Reviewer 3 Report

Comments and Suggestions for Authors

None.